# Effect of Pasteurization by Moderate Intensity Pulsed Electric Fields (PEF) Treatment Compared to Thermal Treatment on Quality Attributes of Fresh Orange Juice

**DOI:** 10.3390/foods11213360

**Published:** 2022-10-25

**Authors:** Rian A. H. Timmermans, Wibke S. U. Roland, Kees van Kekem, Ariette M. Matser, Martinus A. J. S. van Boekel

**Affiliations:** 1Wageningen Food & Biobased Research, Wageningen University & Research, P.O. Box 17, 6700 AA Wageningen, The Netherlands; 2Food Quality and Design, Wageningen University & Research, P.O. Box 17, 6700 AA Wageningen, The Netherlands

**Keywords:** nonthermal processing, minimal processing, pulsed electric fields, ohmic heating, pasteurization, fresh fruit juice, HPLC, GC-MS

## Abstract

Novel pulsed electric field (PEF) process conditions at moderate electric field strength and long pulse duration have recently been established to obtain microbial inactivation. In this study, the effect of these PEF conditions (*E* = 0.9 and 2.7 kV/cm, with pulse duration 1000 µs) at variable maximum temperatures was evaluated on quality attributes of freshly squeezed orange juice. Results were compared to orange juice that received no treatment or a mild or severe thermal pasteurization treatment. No differences for pH and soluble solids were found after application of any treatment, and only small differences were observed for color and vitamin C content (ascorbic acid and dehydroascorbic acid) after processing, mainly for conditions applied at higher temperature. Variations in the maximum temperatures of the PEF and thermal processes led to differences in flavor compounds and the remaining activity of pectinmethylesterase (PME). At PEF conditions with a maximum temperature of 78 °C or higher, PME activity levels were below a critical value, meaning that the cloud is stable. At this temperature volatiles associated with fresh juice (such as octanal and nonanal) are statistically identical to untreated juice, while they are statistically distinguishable from thermal treated. This papers demonstrates the potential of using moderate intensity PEF as an adequate alternative to thermal pasteurization of orange juice with a better retention of the fresh flavor.

## 1. Introduction

Thermal preservation is commonly applied to fruit juices to destroy spoilage and pathogenic micro-organisms as well as to stabilize the juice cloud by deactivating the pectinase enzymes naturally present in the juice. The pH and acidity of a juice play an important role in the growth and activity of micro-organisms and enzymes, and three groups are classified: (1) high-acid foods (pH < 3.7); (2) acid or medium-acid foods (3.7 < pH < 4.5); and (3) low-acid foods (pH > 4.5) [1]. Different pasteurization temperature–time combinations are applied, mainly depending on the acidity of the juice and the inactivation targets. For example, in low-acid juice such as mango, banana, or watermelon the main target is to inactivate pathogenic bacteria, while in medium or high-acid foods such as orange, lemon, or apple the main target is to inactivate spoilage micro-organisms or specific enzymes [2].

Cloud stability is an important quality parameter in orange juice, and can be affected by the impact of pectic degrading enzymes, particularly pectinmethylesterase (PME) [3]. PME de-esterifies the methyl groups on the galacturonic backbone of pectin, creates charged regions that can form complexes with Ca^2+^ and will precipitate, resulting in a clarified juice [3].

Conventional thermal processes for citric juice pasteurization involve the use of temperatures of 65–99 °C and can be categorized as high-temperature short time (HTST) pasteurization (65–86 °C and holding time 1–43 s) and superpasteurization (SP) (90–99 °C and holding times of 1–90 s) [4]. Treatments with temperatures above 100 °C are considered as ultra-high-temperature (UHT) processing. The design of a thermal pasteurization process of orange juice is dependent on the purpose of the treatment. Intense process conditions are required to inactivate PME completely, as PME is thermally more stable than vegetative micro-organisms [5]. Furthermore, the type of juice to be processed plays a role in the required temperature–time combinations. Temperatures and holding times required for the pasteurization of cloudy and pulpy juices are generally higher than those used for clear juice or juice reconstituted from concentrate before aseptic packaging [6].

Conditions at temperatures of 90–95 °C for 15–60 s are reported to be sufficient for microbial and enzymatic inactivation of single-strength juice [7,8]. Braddock reported that conditions at 90–99 °C for 15–30 s are industrially applied [9]. When there is no need for complete PME inactivation, citrus industry usually applies moderate heat treatments of 72 °C for 10–30 s [4].

When orange juice is heated, a complex series of chemical reactions is started involving sugars, amino acids, lipids, ascorbic acid, sulfur-containing components, phenolic compounds, and peel oil components [10]. These reactions can result in a loss of fresh flavor [11], degradation of ascorbic acid [7], and discoloration [12].

Different emerging technologies have been studied and developed over the last decades aiming to inactivate vegetative micro-organisms in a more gentle way at lower temperatures or shorter time compared to a thermal treatment, resulting in a better product quality or lower energy requirement [13,14]. Furthermore, most of the technologies are electricity-driven processing technologies, where the concept of microbial inactivation is based on other principles than only heat, such as pressure, light, sound or electricity [14]. Some technologies are suitable for surface disinfection, while other technologies have a higher penetration depth and result in very rapid volumetric heating. Examples of these technologies include high hydrostatic processing, ultrasound, microwaves, radiofrequency heating, cold plasma, ultraviolet light, ohmic heating, and pulsed electric fields (PEF), but also combinations of technologies are used [15].

PEF processing conditions required to inactivate pathogenic and spoilage micro-organisms in fruit juices are described using variations in the intensity of the pulse (electric field strength, *E*), duration of a single pulse (*τ*), and maximum temperature (*T_max_*) [16]. A recent study of Timmermans et al. [16] distinguished the pulse effects responsible for microbial inactivation from thermal effects, and identified two process windows where pulse effects have an additional effect to the thermal effect responsible for microbial inactivation: (1) *E =* 15–20 kV/cm and *τ* = 2 µs, and (2) *E =* 2.7 kV/cm and *τ* = 1000 µs.

The effect of the first pulse conditions on quality aspects is known and described in previous work [17,18,19]. These studies use *E* > 20 kV/cm and short pulses, and showed a good retention of color, vitamin C content, and flavor when compared to untreated juice; however, no complete inactivation of PME was observed. These conditions are also applied in industry for preservation of juices [20].

The effect of the second pulse condition at *E* = 2.7 kV/cm, *τ* = 1000 µs, and combination with heat on quality aspects is unknown and therefore explored in the present study, in order to evaluate whether these conditions can be a suitable alternative to conventional heat pasteurization, including a complete inactivation of PME, but with better retention of quality.

The aim of the present study was to compare the impact of PEF processing at reduced electric field strength, long pulse duration, and varying maximum temperatures between 45 and 90 °C to two conventional thermal processes, one at mild and another at more intense conditions in freshly squeezed orange juice by studying the quality attributes pH, brix, color, ascorbic acid content, volatile flavor compounds, and PME activity.

## 2. Materials and Methods

### 2.1. The Juice Preparation

Oranges of the variety Salustiana (grown in Spain) were delivered in boxes, stored at 4–6 °C, and divided over two batches (juice A and juice B) before the juice was squeezed on two different days using a commercial juice extractor (Speed pro self-service, Zumex, S.A., Valencia, Spain). The juice was sieved (0.225 mm pore size) to remove the large pulp particles, mixed to make a homogeneous batch to avoid differences between individual oranges, divided and allocated to closed bottles and stored at 2–4 °C prior to processing on the same day.

### 2.2. PEF Processing

A continuous-flow PEF process at lab-scale, described by Timmermans et al. as configuration II [16], was used for electric field treatment. The juice was pumped at a flow rate of 13 ± 1 mL/min, and preheated to 45 ± 1 °C before PEF treatment to reduce the electric input required for the total process. Bipolar square wave pulses with a pulse width (*τ*) of 1000 µs with variable pulse repetition (frequency) were given to heat up the product to a maximum temperature ranging between 45 and 90 °C, with intervals of 3 °C (Table 1). The experiment was performed twice, once with *E =* 2.7 kV/cm (juice A) and the second time at an *E* = 0.9 kV/cm (juice B). The voltage applied by the system was set at 540 V, and two co-linear treatment chambers made of polyetherimide (PEI, Ultem^TM^ resin) with variable dimensions were used to obtain the desired *E*. Table 1 provides the dimensions of the treatment chambers and characteristic parameters of the PEF treatment. The electrodes were made of titanium.

Samples were collected in opposite order: for juice A from the highest temperature possible (75 °C) to the lowest temperature (45 °C), and for juice B from the lowest temperature (45 °C) to the highest temperature (90 °C). Samples were collected after the cooling section and coded with the maximum temperature they received by the PEF-process. An example of a typical temperature-time profile of the PEF treatment is shown in Figure 1I.

Juice with only preheating and no PEF treatment was used as a PEF-control (coded ’s ‘45 °C’) and juice with no preheating nor PEF treatment was a general control (coded as ‘untreated-start’ when taken at the beginning of the day and coded as ‘untreated-end’ when taken at the end of the day).

Pulse waveform, voltage, and current traces were recorded to monitor the intensity in the treatment chambers using a digital oscilloscope (Rigol DS1102E). Temperatures at the inlet and outlet of the treatment chambers were measured with mini-hypodermic thermocouples (HYP-O-T-Type, Omega Engineering, Stamford, CT, USA) and NTC thermistor monitor. The energy input and caloric power measured in the bulk were calculated according to equations given in Timmermans et al. [11]. Specific electrical energy applied by the PEF treatment is given in Table 1, and energy balance between energy input and caloric power measured in the bulk were equal to each other within experimental error (<5%), with absolute deviations of 0.5–3 °C based on maximum temperature.

### 2.3. Thermal Processing

Thermal processing of the orange juices A and B was carried out at continuous-flow conditions at lab-scale. The orange juice was pumped at a flowrate of 22 ± 1 mL/min, preheated to the desired maximum temperature of 72 °C or 95 °C in 43 s by heating through a heat spiral (SS-316, diameter 4 mm) that was immersed in a water bath at 75 °C or 97 °C, followed by a holding-section at the desired temperature in a heat spiral (SS-316, diameter 1 mm) for 20 s, and directly cooled down in a cool spiral (SS-316, diameter 2 mm) immersed in a water bath at 20 °C for 23 s, followed by a cooling section in ice-water (Silicone tube, 1 mm) for 90 s, to obtain a product outlet temperature of about 10 °C. A heating profile of the thermal treatment is shown in Figure 1II. Treatment conditions were a mild thermal treatment (72 °C—20 s) (LH) and a severe thermal treatment (95 °C—20 s) (HH), based on conditions documented to be used in industrial practice [9,21].

### 2.4. Sampling and Storage of Samples

For each treatment condition, five samples per quality attribute were collected in appropriate tubes, and immediately stored on ice after collection. All quality attributes were analyzed in triplicate, by measuring a single analysis per collected sample tube, meaning that for each attribute two spare samples were stored. Samples for flavor analysis were directly prepared in vials, capped and stored at −80 °C. Samples for vitamin C content and enzyme analysis were stored directly in −80 °C as well. Other samples for color, pH, and brix were stored overnight at 4 °C and analyzed the day after production.

### 2.5. Dissolved Oxygen, Conductivity, pH and Soluble Solids

The amount of dissolved oxygen in the orange juice was measured prior to processing, using a handmeter ExStik II DO600 (Extech Instruments, Waltham, USA) at a temperature of 7 °C.

The electrical conductivity of the orange juice was measured with a conductivity meter (Greisinger GMH 3430, Regenstauf, Germany). The pH of the orange juice (19.5 °C ± 1 °C) was measured with a pH meter (827 pH lab, Metrohm, Herisau, Switserland) with a glass electrode (Metrohm) and calibrated before every series of measurement at pH 4 and 7.

The amount of soluble solids of the orange juice samples was measured by a digital hand-held refractometer (PR-1, Atago, Tokyo, Japan) and measured as °Brix at temperatures of 19.5 °C ± 1 °C.

### 2.6. Color Measurements

Color measurements were performed using a Hunterlab ColorFlex spectrophotometer (Hunterlab, Reston, VA, USA). An illuminant of D_65_ and a 10° angle were used as observer. Standardization and measurement of the samples were similar to previous work [12]. CIE *L**, *a**, and *b**-values were measured, representing *L** (lightness), ranging from 0 (black) to 100 (white), *a** quantifying greenness (negative) to redness (positive) and *b** quantifying blueness (negative) to yellowness (positive). Color difference, Δ*E*, was calculated from *L**, *a**, and *b** parameters, using Hunter–Scotfield’s equation (Equation (1)), where *a* = *a*–*a_0_*, *b* = *b*–*b_0_* and *L* = *L*–*L_0_*. The subscript ‘0′ indicates initial color of untreated-start.
Δ*E* = (Δ*a*^2^ + Δ*b*^2^ + Δ*L*^2^)^1/2^(1)
Δ*E* was calculated for each treatment condition, and dependent on the value of Δ*E* the sample was considered to have no noticeable difference (0–0.5), slightly noticeable (0.5–1.5), noticeable (1.5–3.0), well visible (3.0–6.0), or great (6.0–12.0) [17].

### 2.7. Vitamin C Content

The extraction and analysis of the total vitamin C content was measured essentially mainly according to the method described in [18], and is the sum of ascorbic acid (AA) and its oxidized form dehydroascorbic acid (DHAA). Some modifications to the method of Vervoort et al. [18] were made and described below. In short, a dual detection system was used after HPLC separation, by which AA was directly detected by UV, and DHAA indirectly by fluorometric detection after a post-column on-line derivatization with o-phenylenediamine (Sigma Aldrich, Zwijndrecht, The Netherlands).

AA and DHAA were extracted from juice by dilution of 2.5 mL in 1% (*w*/*v*) metaphosphoric acid (Merck, Darmstadt, Germany) with 0.5% (*w*/*v*) oxalic acid dihydrate (Sigma Aldrich), adjusted to pH 2.0 using 10 M NaOH-solution (Sigma Aldrich), to a volume of 2.5 mL. After homogenization, the samples were flushed with nitrogen and centrifuged for 10 min at 3220× *g* at 4 °C. The supernatant was filtered over a 0.2 µm syringe filter and diluted 1:1 with acetonitrile (Actu-All Chemicals, Oss, The Netherlands), to improve the stability of DHAA [22]. An aliquot of 5 µL was used for injection, and vials were stored in dark sample compartment at 4 °C before injection.

Calibration curves were prepared from standard solutions, and composed of AA (10–250 µg/mL) and DHAA (1.25–15 µg/mL) in the extraction buffer, with a total volume of 10 mL. Stock solution of 1 mg/mL AA was made, and DHAA standard was prepared from the AA stock solution as described [18]. Standard solutions were diluted 1:1 with acetonitrile.

The analytical HPLC column used was a Waters symmetry C18 (250 × 4.6 mm, 5 µm particle size) with Spherisorb ODS2 guard cartridge (Waters, Milford, MA, USA). Mobile phase, post-column reagent, and pump conditions were identical to [18]. AA was detected with a UV-detector set at λ_max_ = 263 nm, DHAA was detected by a fluorescence detector set at excitation and emission wavelength 250 and 430 nm, respectively. Run time was 15 min for both analyses and AA was measured at retention time of 7.2 min and DHAA at retention time of 4.1 min.

Reproducibility of AA and DHAA in orange juice samples and standard solutions was evaluated and optimized beforehand. To equilibrate the system, 10 pre-injections of a standard, composed of 30 µg/mL AA and 5 µg/mL DHAA were carried out on a daily basis, followed by the calibration curve and a set of 10 orange juice samples. Measurements of the calibration curve and a set of 10 orange juice samples were repeated for another three times and completed with the calibration curve (so in total 40 samples could be measured per day). The AA and DHAA content was quantified using bracketing calibration, based on peak area of the standards. Orange juice samples were measured over a couple of days in random order. Samples were thawed on the day of analysis to a maximum temperature of 10 °C.

### 2.8. PME Activity

Pectinmethylesterase (PME) activity was quantified by measuring the release of acid during pectin hydrolysis as a function of time at pH 7.0 and temperature of 35 °C, based on [23]. The reaction mixture consisted of 1–14 mL of orange juice sample (amount depended on estimated enzyme activity, with 1 mL for untreated and 14 mL for most intense thermal conditions) and 100 mL of pectin-salt solution (10 g/L citrus pectin (Alfa Aesar, J61021, Thermo Fisher, Kandel, Germany) and 12.35 g/L NaCl in demiwater, stirred for 16 h). The pH of the juice-pectin-salt solution was rapidly brought to pH = 7.1 ± 0.05 using 0.5 M NaOH of 35 °C. During pectin hydrolysis, the pH was maintained constant at pH 7.0 by addition of 0.01 M NaOH using an automatic pH-stat titrator (Metrohm). Minimal measured time was 10 min and maximum measured time was 60 min, dependent of the slope.

Enzyme activity (PEu) was related directly to the amount of NaOH added per minute, as was calculated according to Equation (2) [23].
(2)$$\mathrm{PEu}\;=\frac{\mathrm{ml}\;\mathrm{NaOH}\;\times \;N\;\mathrm{of}\;\mathrm{NaOH}}{\mathrm{ml}\;\mathrm{sample}\;\times \;\mathrm{time}\;\mathrm{in}\;\min}$$

Orange juice samples were measured over a number of days in random order, and the samples to be analyzed were thawed on the day of analysis, to a maximum temperature of 10 °C. Adequate measurements of the same untreated sample at the start of the day, after every three titrations and at the end of the day gave the possibility to correct for small deviations in PEu measured over time and between the two titrators. Therefore, the measured PEu-value was multiplied by a scale factor; this factor was determined by division of the daily average of the untreated samples per titrator by the total average of the untreated samples.

### 2.9. Volatiles

The flavor compounds in the orange juice headspace were analyzed by a combination of solid phase micro extraction (SPME) and gas chromatography mass spectrometry (GC-MS), mainly according to the method described [19]. Prior to the experiments, pre-tests were carried out to select an internal standard, that did not interact with orange juice, showed no overlap with other peaks in the chromatogram, was soluble in water and stayed stable over time in the −80 °C freezer. Menthone showed to meet these criteria. Menthone solution (Sigma Aldrich) was prepared (daily fresh) in water in concentration 1 µL/100 mL water.

The fresh orange juice or just treated orange juice was immediately transferred in a glass vial of 10 mL, filled with 3 mL juice, 3 g NaCl, and 1.5 mL (daily fresh) menthone solution (internal standard), and capped using crimp-seals with a PTFE/silicone septum. The vials were stored in the freezer at −80 °C until GC-analysis.

After controlled thawing to a maximum temperature of 10 °C, vials were homogenized and placed in the cooling tray (7 °C) of the autosampler. The headspace analyses were conducted on a Trace GC (Thermo Finnigan Ultra, Bremen, Germany ) gas chromatograph (GC) coupled to a DSQ-II (Thermo Scientific, Waltham, MA, USA) mass spectrometer (MS) which was equipped with a TriPlus (Thermo Scientific) autosampler. In a first step, each vial was equilibrated in the incubator at 40 °C for 20 min under agitation. Absorption and desorption conditions to a pre-conditioned solid phase micro-extraction (SPME) fiber with 85 µm carboxen/polydimethylsiloxane (CAR/PDMS) absorptive coating (StableFlex, Supelco, Bellefonte, PA, USA) were identical to [19].

Thermal desorption and thermal cleaning of the fiber after each extraction was similar to the protocol of [19], with modification that cleaning was 40 min at 300 °C in this study.

The volatiles were injected in split mode (1/10) and subsequently separated on a ZB-5MS column (30 m × 0.25 mm id, 0.25 µm film thickness, Phenomenex, Torrance, CA, USA). Carrier gas, flow rate, and column oven temperature and time were identical to [19]. Mass spectra were obtained by electron ionization at 70 eV, with a scanning range of 35–400 *m*/*z*. The MS ion source temperature was 230 °C.

The SPME fiber was checked for damage after every run and was replaced when necessary. Damages occurred due to the relatively aggressive nature of some orange juice volatiles. The GC-MS analyses were performed in triplicate, where three sample vials for each tested condition were taken, and the analyses of all samples were performed in random order. The GC-MS total ion chromatograms obtained were evaluated and peak areas were calculated using Xcalibur software (Thermo Fisher, version 2.2). The peak areas were normalized by the peak areas of the internal standard menthone of the individual runs and corrected for the average internal standard peak areas of all juice A and all juice B runs. The reproducibility of the analyses was assessed by calculation of the standard deviation of the normalized peak areas of the respective compounds.

Identification of the peaks was performed by comparing the components’ mass spectra with the reference mass spectra from the NIST Mass Spectral Library (NIST version 2.0). A match factor of >800 was set as limit for identification. However, in some cases, comparison of the mass spectra was not sufficient to unambiguously assign a compound name to a peak. Therefore, measurements of series of alkanes (C_8_–C_20_, Sigma Aldrich) were performed to enable the calculation of an retention index (RI) for each compound. The calculated RI was allowed to deviate maximally by 20 units from RIs found in literature published for the retention on the same column material [24,25,26]. External reference compounds 2-hexenal and octanal were purchased from Merck (Hohenbrunn, Germany), β-myrcene from Janssen Chimica (Beerse, Belgium), α-pinene and β-pinene from Fluka (Buchs, Switzerland), and sabinene, 3-carene, and ocimene from Sigma-Aldrich (Steinheim, Germany), and measured. The retention times and mass spectra of the respective peaks in the orange juice chromatograms were compared to those of the standard compounds to confirm peak identity.

### 2.10. Data Acquisition and Statistical Analysis

Presented results are the mean and standard deviations based on three measurements per quality attribute per treatment condition. Statistical significance for different treatment conditions was estimated by one-way ANOVA for each juice, followed by Tukey’s *post-hoc* test. *p*-values < 0.05 were considered statistically significant. For both juices, statistical differences between the treatments are indicated with a letter, with matching letters showing no significant difference between the samples. All statistical analysis were performed with SPSS version 25 (IBM Corp., Chicago, IL, USA).

## 3. Results

### 3.1. Dissolved Oxygen and Electrical Conductivity

The amount of dissolved oxygen in orange juice was measured after extraction and sieving, before processing, and was 4.0 ± 1.0 ppm at 7 °C. The amount of dissolved oxygen was measured at intervals for five hours, and the value did not change over time. This amount of dissolved oxygen can be considered as low to normal when comparing the values to reported conditions for plant operations, where an amount of 6.5 ppm is considered as normal while an amount of 1.8 ppm represents a commercial deaerated orange juice [27]. As commercial degassing has a larger effect on volatile losses than a pasteurization step [28], no additional degassing step to reduce more oxygen was carried out in this study, as we wanted to examine the effect of different pasteurization processes on flavor compounds. The electrical conductivity was 0.60 S/m at 45 °C for untreated juice A, and 0.53 S/m at 45 °C for untreated juice B.

### 3.2. pH and Soluble Solids

The quality of citrus juice is typically determined by the balance of sweetness and tartness, with sugars and acids as main contributors [29]. The total soluble solids (expressed as °brix) and the acidity or pH of oranges varies for different species, stages in maturity and during the season [30]. The aim in this study was to evaluate if processing, either by moderate intensity PEF and long pulse duration or thermal treatment, changes these values compared to untreated juice. For both juices A and B, the values for soluble solids of PEF-treated and thermal-treated juice were not statistically different from untreated juice, being 11.1 ± 0.1 °Brix for juices A and 11.6 ± 0.1 °Brix for juices B, which can be considered as a normal value [17,31,32].

Moreover, the pH value for juice A and B did not change by thermal or PEF treatment (*p* > 0.05), and values for juices A and B were pH = 3.51 ± 0.01 and pH = 2.99 ± 0.02, respectively. The pH of juice A was comparable to values often reported for orange juice, with a pH typically ranging from 3.4 to 4.3 [17,31,32], so the pH of juice B can be considered as rather low.

The differences in soluble solids and pH between the batches A and B in untreated orange juice might be related to individual differences between oranges. A pre-test carried out with oranges from the same delivery, showed large differences in soluble solids and pH of juice made from individual oranges, ranging between 9.4 and 11.6 °Brix and pH 3.08 and 3.40. It was therefore decided to make one large, pooled batch of orange juice per day to be able to compare only processing effects. Despite the intention of this approach, some differences in soluble solids and pH of the untreated batch A and batch B were obtained, making it difficult to compare the effect of a varying electrical field strength on quality attributes, as the initial quality of untreated juice was different and seemed to play an important role, as will be discussed in the next sections.

Differences in pH and soluble solids caused by the process conditions tested were not expected as samples were evaluated directly after processing. As no shelf life study was carried out, there was no possibility for surviving micro-organisms to start an organic acid fermentation and reduce the pH.

### 3.3. Color

The color of juice is one of the first quality factors that a consumer judges, and has a remarkable influence on its acceptance. Color is also an indicator of the changes that naturally occur in fresh food or of the changes during processing or storage. In the case of fresh orange juice, the typical color is generally a mixture of pigments and carotenoids [33]. Color was measured instrumentally, and Hunter *L**, *a** and *b**-values of the untreated and treated orange juice are shown in Figure 2.

Lightness (*L**-value) was different for the two batches of untreated orange juice, with values of 50.4 ± 0.2 and 48.8 ± 0.1 for batch A and B, respectively. Preheating of the juice to 45 °C (as a control to PEF, without pulsing), already led to a significant increase of the *L**-value for about 1 point. Further temperature increment by the PEF or thermal process slightly increased the *L**-value for juice A, whereas this trend was not observed for juice B. These small changes are in agreement with earlier results [12] showing an increase in *L**-value after pasteurization of orange juice, which might be attributed to partial precipitation of unstable, suspended particles in the juice [34]. This small difference in *L**-value might (also) be introduced by the pumping of the orange juice, as the difference in *L**-value was observed in the 45 °C samples as well.

Redness (*a**-value) was different for the two batches of untreated orange juice, with values of 3.7 ± 0.2 for batch A and 2.5 ± 0.2 for batch B. Moderate intensity PEF and thermal processing of the juice did not change the *a**-value for juice A, while a slight decrease of this *a**-value (reduction of the redness) was visible for PEF and thermal treatment of juice B (*p* < 0.05).

The values for yellowness (*b**-value) for the two batches were comparable for untreated juice, with values of 60.3 ± 0.3 for juice A and 59.9 ± 0.2 for juice B. However, the batches responded differently to the treatments applied. Preheating of juice A to a temperature of 45 °C (PEF-control) increased the *b**-value with 1 point, but further temperature increase with the PEF treatment did not change the *b**-value. Thermal treatment led to a further increase of the *b*-*value of juice A. Contrarily, preheating to 45 °C led to a decrease in *b** value for juice B. An increase in the *b*-*value of PEF-treated samples was visible at maximum temperatures above 75 °C, but this temperature effect was not visible in thermal-treated juice.

Results of the increase of the temperature dependent increase of the *b**-value are in agreement with earlier results [12] showing an increase of the *b**-value after pasteurization as well.

Although small deviations and trends for the juices in specific *L*, a*,* and *b**-values were observed, it is rather difficult to draw conclusions from these individual values on the overall impact of the processing on color. It is more valuable to compare the total change in *L*, a*,* and *b**-values, expressed in Δ*E* values, as consumers do not judge each particular attribute, but the combination of them [17]. The Δ*E* values were calculated according to equation 1, and results are shown in Figure 2. Preheating of orange juice to 45 °C (PEF control) gave a slightly noticeable change in color for both juice A and juice B. Further temperature increment of the PEF treatment led to higher values of Δ*E*, which is mostly considered as a ‘slightly noticeable’ effect. Only maximum temperatures of PEF between 72 °C–78 °C were characterized as ‘noticeable’ different. Further temperature increment as given by the thermal treatments showed a more ‘noticeable’ difference for juice A, where this was reduced for juice B. It can therefore be concluded that the application of the moderate intensity PEF treatment gave a (slightly) noticeable effect on color compared to untreated juice, while application of a thermal treatment would lead to larger, noticeable effects, meaning that moderate intensity PEF is more gentle than a heat treatment.

### 3.4. Vitamin C

Vitamin C content is an important attribute to orange juice. Although is it not the only fruit product containing vitamin C, it is certainly an important source, as the vitamin C content in orange juice is high, as is the consumption rate of orange juice by humans [35]. Variations in vitamin C content can be found in different citrus products due to factors such as variety, maturity, and cultural practices of the fruit [30], but also due to processing practices and storage conditions of these products before they reach the consumer [35].

The vitamin C content was determined by measuring the concentration of ascorbic acid (AA) and dehydroascorbic acid (DHAA), together forming the vitamin C content, and values of untreated and processed orange juice for AA and DHAA are shown in Figure 3.

AA content for untreated juice was 511.5 ± 22.0 mg/L for juice A and 579.3 ± 20.9 mg/L for juice B, and DHAA content was 8.1 ± 0.7 mg/L for juice A and 6.6 ± 0.4 mg/L for juice B. According to various references, this orange juice can be considered as a rich source of vitamin C [18,31,35]. Comparison of the amount of AA present in the samples before and after pasteurization (either by the moderate intensity PEF process or thermal treatment) did not show any significant change introduced by the pasteurization process applied (Figure 3I). Probably the low pH of the juice and low amount of dissolved oxygen were responsible for this AA retention, as these factors are known to be important factors to stabilize the AA [36].

Contrarily, significant changes were detected for DHAA, which concentration increased when treatment temperature increased, either applied by the PEF process or by the thermal treatment (Figure 3II). DHAA is the first oxidation product of AA, and its levels can increase when exposed to high temperatures, light or oxygen, neutral pH, oxidases as well as to the presence of some traces of heavy metal ions. DHAA still exhibits vitamin C activity, since it can be reconverted into AA in the human body [36]. However, this compound is unstable and most of it is lost as diketogulonic acid, which has no vitamin C activity [36].

The concentration of DHAA formed in our study was rather low compared to the concentration of AA. Reduction in the AA concentration was therefore not noticed within experimental error, as the standard deviations in AA measurements were larger than the concentrations of DHAA formed. The total amount of vitamin C, being the sum of AA and DHAA, did not change by any process or specific process condition used, either by moderate intensity PEF treatment or thermal treatment (*p* > 0.05).

### 3.5. PME

Pectinmethylesterase (PME) is generally responsible for cloud loss of orange juice by de-esterification of pectin [3], and it is therefore important to completely inactivate this enzyme to make shelf life stable products. The remaining enzyme activity of PME after the different treatments was determined, and results are shown in Figure 4. The activity unit of PME (expressed as PEu) is defined as the amount of enzyme that releases 1 µmole of carboxylic acid groups in 1 mL of solution for one minute.

Processing clearly reduced enzyme activity, where higher maximum temperatures of the moderate intensity PEF or thermal process led to more reduction of the PME activity. A difference in PME activity between the two PEF processes and batches A and B was observed, mainly in the PEF-treated samples with conditions up to a maximum temperature of 60 °C. PEF conditions applied with maximum temperature of 63–75 °C showed no significant difference between the processes and batches A and B, and no significant difference between the batches A and B were observed for the thermal treatments applied (this is not indicated with indices in Figure 4). The slight difference in pH content between juice A (pH = 3.5) and B (pH = 3.0) might play a role here, as pH plays an important role in the temperature requirements to inactivate PME. When the pH of juice was decreased, the susceptibility of the PME enzyme to heat inactivation increased [8,37]), leading to more inactivation at equal maximum temperatures. Adjustment of the pH of the juice to values below pH = 3.5 showed a reduction of 20–40%, even at temperatures as low as 50 °C [38]. In addition to the pH, differences between batches A and B might also be explained by different PME isoenzymes present. Orange juice comprises multiple forms of pectinesterase, known as isoenzymes, with different kinetic properties [3], that can generally be classified into heat-labile PME and heat-stable PME. Typically, orange juice consists of 90% of the heat-labile PME, which is destroyed at temperatures below or near 70 °C, and 10% of the heat-stable PME that is destroyed around temperature of 90 °C or higher, dependent on holding times used [8,37,39]. Based on the PME activity shown in Figure 4, juices A and B contained the same degree of heat-stable PME fraction, being 11.1 % at the maximal activity at a temperature of 69 °C for both juices. Thus, the thermo-labile fraction of juice B treated at 0.9 kV/cm was more temperature-sensitive than thermo-labile fraction of juice A treated at 2.7 kV/cm, which might have been caused by the effect of pH.

Juice is considered commercially stable when the remaining PME activity is smaller than 1 × 10^−4^ PEu/mL [9,37,40]. This threshold value is indicated in Figure 4 with a dashed line to facilitate an easier comparison. The results of this study showed that mild thermal treatment did not meet this specification, but intense thermal treatment did meet this criterion, as expected on forehand based on the temperature-time requirements reported for the thermal inactivation of the resistant PME fraction [8]. More interesting is that moderate intensity PEF treatments with long pulse duration at maximum temperatures of 78 °C or higher meet this criterium as well. Although this temperature of 78 °C is much lower than the intense thermal treatment applied at 95 °C, we cannot quantify if the moderate intensity PEF treatment has a beneficial ‘pulse or electroporation effect’, or whether it is ‘only’ a thermal effect, as only two thermal treatment conditions were evaluated.

PEF studies carried out using high intensity pulses claim that enzyme inactivation is predominantly caused by thermal effects (>90%), rather than by the voltage pulses themselves [41,42], which seems to be confirmed in the present study using pulses with moderate intensity.

### 3.6. Volatiles

Flavor is one of the most important attributes of orange juice, and processing is known to irreversibly change the ‘fresh-like’ attributes of the juice in a negative way [9]. Early research showed that the unique flavor of orange juice could be ascribed to a couple of specific aldehydes, but extensive analytical studies during the past decades showed that a mixture of several compounds in the proper proportions and concentrations is necessary for a good flavor [43]. The perceived flavor is a combination of volatile aroma compounds and non-volatile taste compounds that can interact with each other and with the matrix. Processing of orange juice will affect the volatile aroma compounds much more than the non-volatile taste compounds. Therefore, the focus of the current study lies in the investigation of the relative amounts of volatile compounds and their changes upon processing.

The volatile compounds from the headspace of the orange juice were extracted by solid-phase micro extraction (SPME) and subsequently separated with gas chromatography (GC) and identified by mass spectrometry (MS). A total of 34 compounds was analyzed in this study. A typical chromatogram is shown in Figure 5.

A relatively low number of esters, aldehydes, and alcohols were detected in juice A and juice B, no ketones, ethers, and acids were detected at all, whereas a relative high number of terpenes was detected in the juice when compared to the literature [43].

Retention index values of all compounds were determined and are given in Table 2. A total of 32 compounds were identified according to the criteria mentioned above. Two peaks could not be identified, and one compound has been tentatively identified as β-phellandrene, as the match factor of the MS fragmentation spectra and the calculated RI deviated slightly from the criteria described above. 

For all compounds, the areas under the peaks were calculated and normalized according to the internal standard. For each compound, the areas under the peaks were compared for all processing conditions.

In general, relatively small changes in the peak area for the different compounds were introduced by processing. Six different trends in relative flavor concentrations were observed and statistically quantified when the effect of processing on individual compounds was tested. Examples to illustrate the six trends are shown in Figure 6.

Figure 6A illustrates a trend where the compound 1-octanol in juice B seemed to be unaffected by processing (untreated = PEF = heat). Figure 6B illustrates the trend with higher concentrations of β-linalool in untreated and PEF-treated than in heat-treated juice B (untreated = PEF > heat). Figure 6C illustrates the trend with higher concentrations of limonene in untreated than in PEF and heat-treated juice B (untreated > PEF = heat). Figure 6D illustrates the trend with lower concentrations of octanal in PEF-treated juice A than in untreated juice A, and even lower concentrations after thermal treatment (untreated > PEF > heat). Figure 6E illustrates equal concentrations of 4-terpineol in untreated juice B and PEF at reduced temperature, but higher concentrations at PEF at higher temperature and in thermal treated (untreated = PEF_<72 °C_ ≤ PEF_≥72 °C_ = heat). Figure 6F illustrates lower concentrations of 4-terpineol in untreated and PEF-treated juice A than in thermally treated (untreated = PEF < heat).

Due to analytical inconsistencies, the GC-MS analyses for A-LH, B-60, B-63, B-75, and B-87 had to be excluded. However, an adequate number of GC-MS analyses remained in order to compare untreated orange juice to orange juice treated at various temperatures with moderate intensity PEF and with conventional heat treatment.

The trends for each compound are quantified and given in Table 2, and figures including normalized peak areas per compound and treatment and statistical analysis are provided in the Appendix A. The aroma compounds that have been reported in the literature to contribute directly or indirectly to the overall orange juice flavor are indicated in Table 2, and the impact of processing on (only) these compounds will now be discussed.

No impact of processing with moderate intensity PEF or thermal processing was statistically quantified when the measured peak areas were compared to untreated (U = PEF = heat) (*p* > 0.05) in both juices for ethyl acetate, ethyl butyrate, 1-octanol, neral, geranial, and in juice A for α-terpinene, β-linalool, nonanal, and α-terpineol.

Processing of the juice by either a moderate intensity PEF treatment or thermal treatment reduced the peak area compared to untreated juice (U > PEF = heat) for both juices for component α-pinene, β-pinene, and limonene, in juice A for decanal.

However, for this study, it was more interesting to evaluate whether a difference between the moderate intensity PEF treatment and thermal treatment was observed (PEF > heat). Thermal treatment reduced the relative concentration of the positive flavor contributors β-myrcene (juice A and B) and β-linalool (juice B) significantly, whereas the area of the moderate intensity PEF treatment did not change compared to untreated juice (U = PEF > heat). A beneficial effect for the moderate intensity PEF treatment compared to the thermal treatment was also observed for octanal (both juices), α-pinene (juice A), nonanal, and decanal (juice B), although the areas for these moderate intensity PEF-treated compounds were lower compared to values in untreated juice (U > PEF > heat). For nonanal (juice B), a differentiation within the PEF group was observed, where at PEF temperatures above 78 °C less of the component was detected.

PEF and thermal treatment could also lead to the formation of components, such as α-terpinene, 4-terpineol, and α-terpineol. These latter two compounds are typical reaction products of acid-catalyzed degradation of limonene and β-linalool, where the rate of formation is dependent on pH of the juice [46] or formed at high temperatures [11] and may be recognized as compounds causing off-flavor when present in high concentrations [11,46]. A temperature-dependent increase of both compounds was found in PEF-treated juice, when the maximum temperatures of the treatment was 72 °C or higher (α-terpineol and 4-terpineol) (juice B). Interestingly, no effects in PEF-treated juice A were observed, although this juice was measured up to maximum temperatures of 75 °C. Thermal processing led to an increase of these off-flavor compounds, to a similar extent as the most intense PEF treatments tested.

As all the data were normalized, it is not possible to speculate if the differences that are observed by any treatment will also be perceived.

Our results using moderate intensity PEF are showing a lot of similarities with research of others [47] who compared ohmic heating with conventional heat treatment of orange juice, demonstrating a higher concentration of pinene, myrcene, octanal, limonene, and decanal in ohmic heat-treated than in conventional heated orange juice. Another study [48] showed a better retention of concentration of α-pinene, myrcene, octanal, limonene, and decanal after high intensity PEF treatment at *E* = 40 kV/cm than after a conventional heat treatment. PEF treatments at *E* = 20 kV/cm, *τ* = 25 µs, and energy input of 100 or 150 kJ/kg showed better retention of limonene, β-myrcene, α-pinene, and valencene compared to thermal treatment at 95 °C—30 s [49]. All studies described in the literature, as well as our study, had a lower heat load for the PEF or ohmic treatment compared to the conventional heat treatment, probably leading to the initiation of fewer chemical reactions, and therefore resulting in more retention of flavor compounds.

Overall, it can be said that the impact of the thermal treatments on the volatile flavor compounds was very moderate, and only at PEF processing temperatures of 72 °C or higher minimal changes were observed. Some deviations between moderate intensity PEF-treated and thermally processed orange juice were found for individual compounds, where moderate intensity PEF treatment resulted in a better retention of flavor compounds. No strong off-flavor compounds such as *p-*cymene and carvone were detected, such as in other studied juices [11], and off-flavor compounds such as 4-vinylguaiacol and furaneol could not be detected as they need a solvent extraction [50]. The increase of α-terpinene, 4-terpineol, and α-terpineol was minimal at the most intense PEF and thermal treatments studied, when compared to the concentrations in untreated juice.

## 4. Conclusions

In this paper we compared the effect of PEF processing at reduced electric field strength (*E* = 0.9 and 2.7 kV/cm) and long pulse duration (*τ* = 1000 µs) in combination with varying maximum temperature to two conventional thermal pasteurization processes on several quality aspects in orange juice. The initial quality of the two batches of untreated juices A and B showed some variations. Furthermore, the juices had a relatively low pH, so some chemical reactions that could influence the quality attributes ascorbic acid content, color and some flavor compounds, might be lower than expected beforehand and could be more pronounced in other products with higher pH. Taking this in account, we concluded from this study: The pH and soluble solids did not show a difference after any treatment either by moderate intensity PEF or conventional heat. Only small differences were observed for color and vitamin C content, where an increased temperature impact led to slight deviations, mainly for the most intense thermal treatment.

A real impact of processing was measured in the enzymatic activity of PME, which reduced when the temperature was increased. Reduction of the remaining PME activity to levels below 1.0 × 10^4^ PEu were reached with moderate intensity PEF processing at maximum temperatures of 78 °C or higher. According to the literature, this level is sufficient to obtain a shelf stable orange juice. The impact of processing on the flavor components showed that the beneficial contributors to the flavor were better retained after moderate intensity PEF treatment than after the conventional thermal treatment, and that the relative concentration of compounds with a negative flavor impact was similar at most intense PEF conditions of 90 °C to the intense conventional heating, and that no compounds causing strong off-flavors were detected.

Larger differences in quality between moderate intensity PEF processing and thermal processing could also be expected when experiments would be performed on larger pilot- or industrial scale. The heat transfer for PEF treatment is scale independent due to volumetric heating, but for the conventional thermal process a lower heat transfer will be expected at larger scale than on the lab-scale unit, leading to longer process times, which may affect product quality negatively.

Comparison of the effect of the two electric field strengths used in this study (*E =* 0.9 kV/cm and 2.7 kV/cm with duration of *τ* = 1000 µs) did not show remarkable differences on quality attributes, while identical pulse conditions tested did show a significant higher inactivation of micro-organisms at an electric field strength of 2.7 kV/cm compared to 0.9 kV/cm [16]. PEF processing conditions at moderate intensity (*E =* 2.7 kV/cm), long pulse duration (τ = 1000 µs) and combination with heat up to maximum temperatures of 65–90 °C led to more microbial and enzymatic inactivation than high intensity PEF conditions at *E =* 20 kV/cm, short pulse duration (2 µs) and lower maximum temperatures (T < 60 °C) [16,17,18,19], and therefore a longer shelf life using the moderate intensity PEF processing conditions could be expected, although this remains to be determined. Another suggestion for future work is to discriminate thermal effects from electric effects during the PEF-treatment.

Based on the results of this study, PEF processing at moderate electric field strength, long pulse duration at a maximum temperature of 78 °C would be an optimal process condition, which is suitable as alternative pasteurization process to thermal treatment for low pH orange juice, as the PME level is reduced to the desired number, microbial inactivation criteria could be met [16] and quality aspects color, vitamin C content and flavors are better retained than after thermal treatment, being still very close to untreated juice.

## Figures and Tables

**Figure 1 foods-11-03360-f001:**
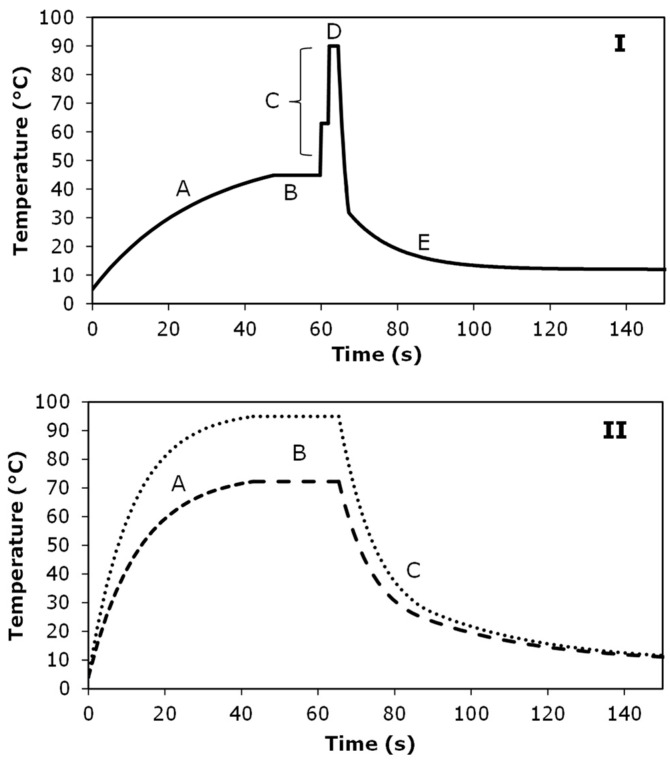
Example of temperature-time profile of the moderate intensity PEF treatment (**I**) and thermal treatment (**II**) of orange juice. Panel (**I**). Orange juice was preheated from 4 °C to 45 °C in 48 s (A), maintained at 45 °C for 12 s (B) before entering the treatment chambers, heated up in the first treatment chamber, followed by a pause of 1.7 s, the time to go to the second treatment chamber, and heated up to a desired maximum temperature, i.e., 90 °C (C). Then orange juice is transferred to cooling section in 2.3 s (D) and cooled down for 122 s to temperatures < 10 °C ©. Panel (**II**)**.** Orange juice was preheated from 4 °C to 72 °C for low thermal treatment or to 95 °C for high thermal treatment in 43 s (A), maintained at the desired temperature for 20 s (B) and cooled down for 113 s to temperatures < 10 °C (C).

**Figure 2 foods-11-03360-f002:**
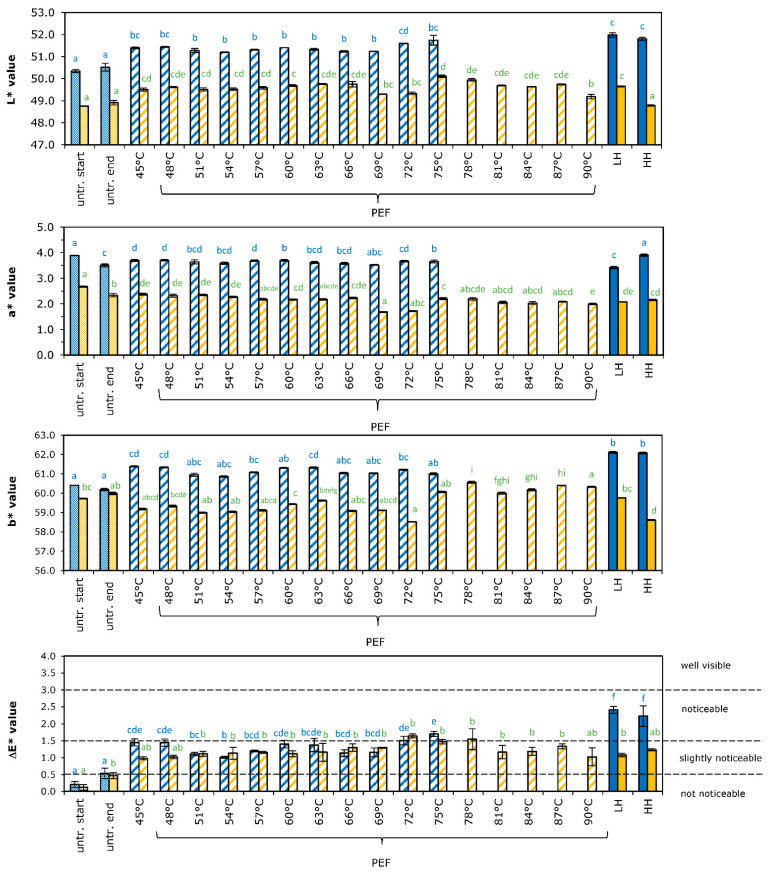
Effect of PEF and thermal processing on color of orange juice, according to Hunterlabs *L**, *a**, and *b**-value and total color difference (Δ*E*). Blue bars represent juice A, orange bars represent juice B. Values of untreated juice at start and end of the day are given. Results of various moderate intensity PEF treatment are given according to the maximum temperatures applied, using an *E =* 2.7 kV/cm (blue) and 0.9 kV/cm (orange). Thermal treatments are indicated as LH for low heat treatment (75 °C—20 s) and HH for high heat treatment (95 °C—20 s). Bars represent the average of three measurements with standard deviation. Statistical differences between the treatments are indicated with a letter per juice (color), with matching letters showing no significant difference.

**Figure 3 foods-11-03360-f003:**
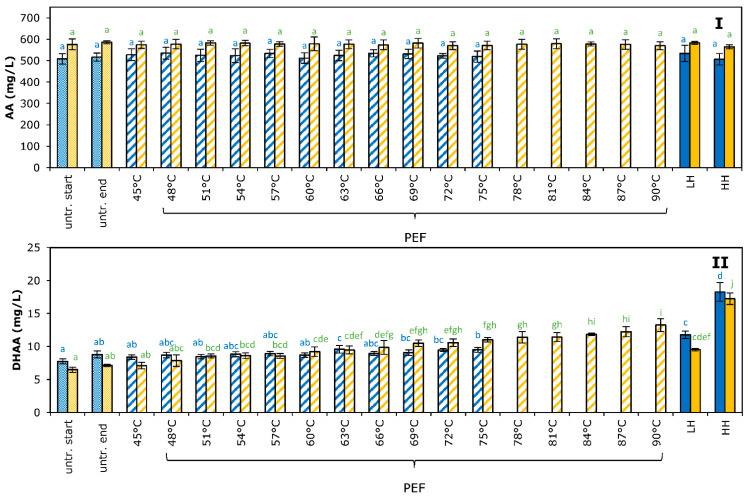
Effect of PEF and thermal processing on ascorbic acid (AA) (panel (**I**)) and dehydroascorbic acid (DHAA) (panel (**II**)) concentration in orange juice. Blue bars represent juice A, orange bars represent juice B. Values of untreated juice at start and end of the day are given. Results of various moderate intensity PEF treatment are given according to the maximum temperatures applied, using an *E* = 2.7 kV/cm (blue bars) and 0.9 kV/cm (orange bars). Thermal treatments are indicated as LH for low heat treatment (75 °C—20 s) and HH for high heat treatment (95 °C—20 s). Bars represent the average of three measurements with standard deviation. Statistical differences between the treatments are indicated with a letter per juice (color), with matching letters showing no significant difference.

**Figure 4 foods-11-03360-f004:**
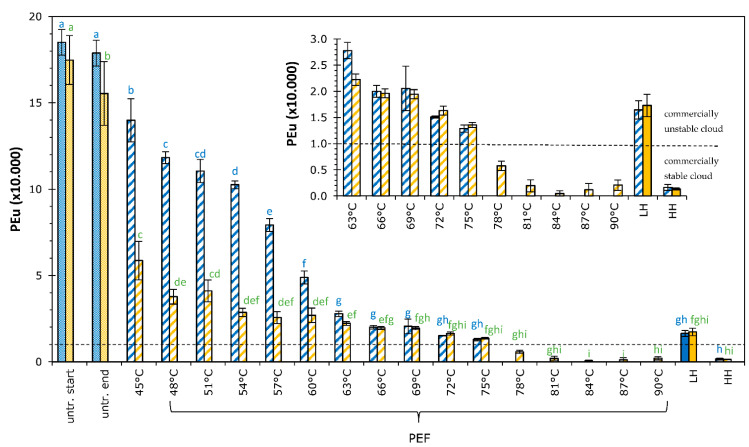
Effect of PEF and thermal processing on remaining pectinmethylesterase activity, expressed as pectinesterase units (PEu) in orange juice. Blue bars represent juice A, orange bars represent juice B. Values of untreated juice at start and end of the day are given. Results of various moderate intensity PEF treatment are given according to the maximum temperatures applied, using an *E =* 2.7 kV/cm (blue bars) and 0.9 kV/cm (orange bars). Thermal treatments are indicated as LH for low heat treatment (75 °C–20 s) and HH for high heat treatment (95 °C–20 s). Bars represent the average of three measurements with standard deviation. Statistical differences between the treatments are indicated with a letter per juice (color), with matching letters showing no significant difference. Juice cloud is considered commercially stable when PEu values are below 1.0 × 10^−4^ (indicated with dashed line).

**Figure 5 foods-11-03360-f005:**
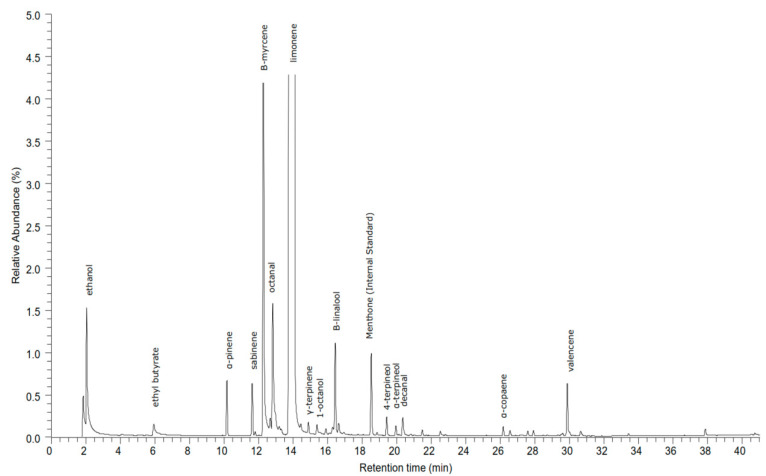
Typical GC-MS chromatogram of orange juice flavor compounds.

**Figure 6 foods-11-03360-f006:**
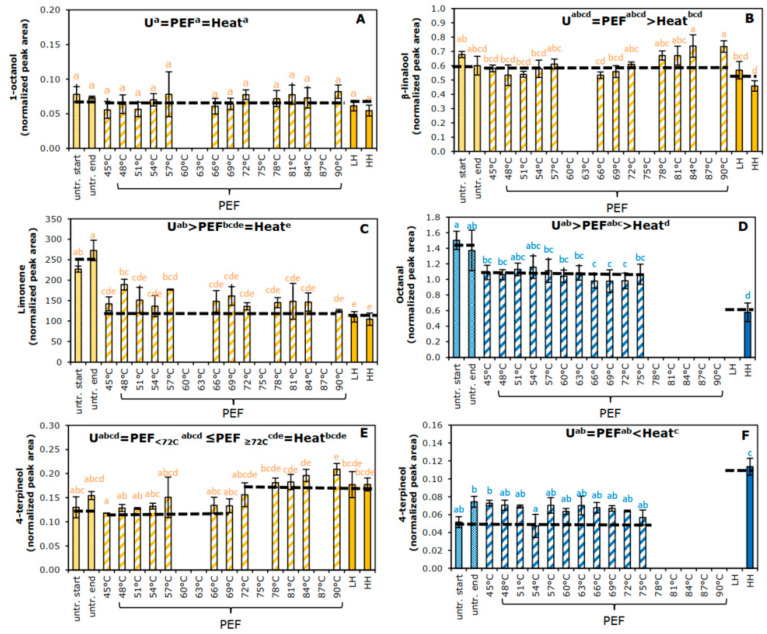
Trends illustrating the effect of processing on the detected volatiles present in orange juice, discriminating untreated juice (U), moderate intensity PEF-treated juice (PEF), and thermal-treated juice (heat). Six trends were identified, one trend showed no processing effect: U = PEF = heat (shown in (**A**) for 1-octanol), three trends showed a decrease of the compound after PEF and/or thermal processing compared to untreated, being U = PEF > heat (shown in (**B**) for β-linalool), U > PEF = heat (shown in (**C**) for limonene), and U > PEF > heat (shown in (**D)** for octanal); and two trends showing an increase of the compound after PEF and/or thermal processing compared to untreated: U = PEF_low_ < PEF_high_ = heat (shown in (**E**) for 4-terpineol) and U = PEF < heat (shown in (**F**) for 4-terpineol). Bars represent the average of three measurements with standard deviation. Trends for each specific compound are shown in the Appendix A, and summarized in Table 2. Statistical differences between the treatments are indicated with a letter per juice, with matching letters showing no significant difference.

**Table 1 foods-11-03360-t001:** Characterizations of the pulse conditions and configurations used to apply a moderate intensity PEF treatment at a flow rate of 13 mL/min.

Juice	*E* (kV/cm)	*τ* (µs)	Treatment Chamber (Diameter × length) (mm)	Residence Time (ms)	Pulse Repetition Frequency (Hz)	Number of Pulses Applied	Specific Electric Energy (kJ/kg)	Tmax (°C)
A	2.7	1000	1.0 × 2.0	14	0–112	0–3.2	0–114	45–75
B	0.9	1000	4.0 × 6.0	696	0–48	0–67	0–171	45–90

**Table 2 foods-11-03360-t002:** Volatile compounds in treated and untreated orange juice.

No.	RI	Compound Name	Class	Identification ^§^	Trends in Juice A *	Trends in Juice B *	Impact on Orange Juice Flavor
1	<800	Ethanol	Alcohol	MS	U ^a^ = PEF ^a^ = Heat ^a^	U ^a^ = PEF ^a^ = Heat ^a^	
2	<800	Ethyl acetate	Ester	MS	U ^a^ = PEF ^a^ = Heat ^a^	U ^a^ = PEF ^a^ = Heat ^a^	Background aroma, contributing to naturalness ^†^
3	802	Ethyl butyrate	Ester	MS, RI	U ^a^ = PEF ^a^ = Heat ^a^	U ^a^ = PEF ^a^ = Heat ^a^	Key odorant in fresh orange juice^†^
4	863	2-Hexenal	Aldehyde	MS, RI, St.	U ^a^ = PEF ^a^ = Heat ^a^	n.d.	
5	932	α-Pinene	Monoterpene	MS, RI, St.	U ^a^^b^ > PEF ^abc^ = Heat ^c^	U ^a^^b^ > PEF ^bcdef^ = Heat ^ef^	Background aroma, contributing to naturalness ^†^
6	972	Sabinene	Monoterpene	MS, RI, St.	U ^a^^b^ > PEF ^abcd^ > Heat ^d^	U ^a^ > PEF^bcde^ > Heat ^ef^	
7	977	β-Pinene	Monoterpene	MS, RI, St.	U ^a^^b^ > PEF ^abc^ = Heat ^c^	U ^a^^b^ > PEF^bcd^ = Heat ^d^	Background aroma, contributing to naturalness ^†^
8	990	β-Myrcene	Monoterpene	MS, RI, St.	U ^a^^b^ > PEF ^abcd^ > Heat ^d^	U ^a^^b^ > PEF ^abcd^ ≥ Heat ^d^	Background aroma, contributing to naturalness ^†^
9	1002	β-Phellandrene	Monoterpene	tentative	U ^a^ = PEF ^a^ = Heat ^a^	U ^a^ = PEF ^a^ = Heat ^a^	
10	1005	Octanal	Aldehyde	MS, RI, St.	U ^a^^b^ > PEF ^abc^ > Heat ^d^	U ^a^^b^ ≥ PEF ^abc^ ≥ Heat ^c^	Key odorant in fresh orange juice ^†^
11	1008	3-carene	Monoterpene	MS, RI, St.	U ^a^^b^ > PEF ^abc^ = Heat ^c^	U ^a^^b^ > PEF ^abcd^ = Heat ^cd^	
12	1014	*Unidentified*		-	U ^a^ = PEF ^a^ = Heat ^a^	U ^a^ = PEF ^a^ = Heat ^a^	
13	1017	α-Terpinene	Monoterpene	MS, RI	U ^a^ = PEF ^a^ = Heat ^a^	U ^a^^b^ = PEF ^ab^ < Heat ^ab^	Background aroma, contributing to naturalness ^†^
14	1036	Limonene	Monoterpene	MS, RI	U ^a^^b^ > PEF ^abc^ = Heat ^c^	U ^a^^b^ > PEF ^bcde^ = Heat ^e^	Necessary for aroma, function uncertain ^†^
15	1048	Ocimene	Monoterpene	MS, RI, St.	U ^a^ = PEF ^a^ = Heat ^a^	U ^a^^b^ > PEF ^bcd^ = Heat ^cd^	
16	1059	γ-Terpinene	Monoterpene	MS, RI	U ^ab^ = PEF ^ab^ ≤ Heat ^b^	U ^a^ = PEF ^a^ = Heat ^a^	
17	1073	1-Octanol	Alcohol	MS, RI	U ^a^ = PEF ^a^ = Heat ^a^	U ^a^ = PEF ^a^ = Heat ^a^	Background aroma, contributing to naturalness ^†^
18	1086	Terpinolene	Monoterpene	MS, RI	U ^a^^b^ > PEF ^ab^ = Heat ^ab^	U ^a^^b^ > PEF ^abc^ = Heat ^abc^	
19	1099	β-Linalool	Monoterpene alcohol	MS, RI	U ^a^ = PEF ^a^ = Heat ^a^	U ^ab^^cd^ = PEF ^abcd^ > Heat ^bcd^	Key odorant in fresh orange juice ^†^
20	1105	Nonanal	Aldehyde	MS, RI	U ^a^ = PEF ^a^ = Heat ^a^	U ^a^^b^ ≥ PEF_<78C_ ^abc^ ≥ PEF_≥78C_ ^bc^ = Heat ^c^	Key odorant in fresh orange juice ^†^
21	1182	4-Terpineol	Monoterpene alcohol	MS, RI	U ^ab^ = PEF ^ab^ < Heat ^c^	U ^abc^^d^ = PEF_<72C_ ^ab^^cd^ ≤ PEF_≥72C_ ^abcde^ = Heat ^bcde^	Background aroma, contributing to naturalness ^†^,
							marker for heat-abuse, off-flavor at high conc. ^‡^
22	1196	α-Terpineol	Monoterpene alcohol	MS, RI	U ^a^ = PEF ^a^ = Heat ^a^	U ^ab^^c^ = PEF_<72C_ ^a^^b^ ≤ PEF_≥72C_ ^abc^ = Heat ^abc^	Marker for heat-abuse, off-flavor at high conc. #
23	1207	Decanal	Aldehyde	MS, RI	U ^a^^b^ > PEF ^ab^ = Heat ^b^	U ^ab^ > PEF ^abcd^ ≥ Heat ^cd^	Key odorant in fresh orange juice ^†^
24	1239	Neral	Monoterpene aldehyde	MS, RI	U ^a^ = PEF ^a^ = Heat ^a^	U ^a^ = PEF ^a^ = Heat ^a^	Key odorant in fresh orange juice ^†^
25	1267	Geranial	Monoterpene aldehyde	MS, RI	U ^a^ = PEF ^a^ = Heat ^a^	U ^a^ = PEF ^a^ = Heat ^a^	Key odorant in fresh orange juice ^†^
26	1377	α-Copaene	Sesquiterpene	MS, RI	U ^a^ = PEF ^a^ = Heat ^a^	U ^a^ > PEF ^abcd^ > Heat ^d^	
27	1389	β-Cubebene	Sesquiterpene	MS, RI	U ^a^ > PEF ^ab^ > Heat ^b^	U ^ab^ > PEF ^abcde^ ≥ Heat ^de^	
28	1408	Dodecanal	Aldehyde	MS, RI	n.d.	n.d.	Background aroma, contributing to naturalness ^†^
29	1420	β-Caryophyllene	Sesquiterpene	MS, RI	U ^a^ > PEF ^ab^ > Heat ^b^	U ^a^ > PEF ^abcde^ ≥ Heat ^de^	
30	1431	*Unidentified*	Sesquiterpene	-	U ^a^^b^ > PEF ^abc^ > Heat ^c^	U ^a^ > PEF ^abcde^ ≥ Heat ^cde^	
31	1456	α-Caryophyllene	Sesquiterpene	MS, RI	U ^a^ = PEF ^a^ = Heat ^a^	U ^a^^b^ > PEF ^bc^ = Heat ^c^	
32	1484	γ-Selinene	Sesquiterpene	MS, RI	U ^a^ = PEF ^a^ = Heat ^a^	U ^abc^ = PEF ^abcde^ ≥ Heat ^e^	
33	1493	Valencene	Sesquiterpene	MS, RI	U ^a^ = PEF ^a^ = Heat ^a^	U ^ab^ = PEF ^ab^ = Heat ^ab^	
34	1522	δ-Cadinene	Sesquiterpene	MS, RI	U ^a^ = PEF ^a^ = Heat ^a^	U ^a^ = PEF ^a^ = Heat ^a^	

^§^ MS: Identification based on MS fragmentation spectra in NIST compound library RI: Identification based on retention indices found in the literature. St: Identification based on retention time and MS spectra of an external reference compound. * Trends between untreated, PEF-treated, and heat-treated orange juice as indicated in Figure 6, with letters identifying the groups that are classified based on *Posthoc* analysis. a–f, Statistical differences between the treatments are indicated with a letter per juice. Trends between untreated, PEF, and heat are indicated per group with <, ≤, =, >, and ≥. Underlined letters are dominant for the specific treatment. For details of the individual components consult the Appendix A. n.d. not determined. † Impact on fresh orange juice flavor reported in the review of [43], including only compounds that have been shown to be aroma active in GC-Olfactometry studies and that have been identified as relevant for orange juice flavor by two or more independent research groups. ‡ Impact on processed orange juice, from GC-Olfactometry studies, according to [11,44]. **#** Impact on processed orange juice, from GC-Olfactometry studies, according to [11,45].

## Data Availability

The data used to support the findings of this study can be made available by the corresponding author upon request.

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
