# Peer review of "Effect of Pasteurization by Moderate Intensity Pulsed Electric Fields (PEF) Treatment Compared to Thermal Treatment on Quality Attributes of Fresh Orange Juice"

_foods, 2022, doi:10.3390/foods11213360_

Round 1

Reviewer 1 Report

The manuscript is related to an interesting application on mild technology to improve the quality of stabilized orange juices. In the introduction where the authors describe the thermal treatments could be applied should remind the influence of the pH since if we are generally referring to "fruit juices" and nnot only to citrus products the pH can be different and thus influencing the thermal treatment effect. The experimental design related to conventional and PEF assisted pasteurization might consider a comparison on F0 values computed from the thermal profiles and to use those values to compare the effect on quality parameters as a function of the different F0 values. It could be very useful to give an explanation of the difference in quality due to the applied thermal treatments, with and without PEF treatment 

Author Response

Comments and Suggestions for Authors

The manuscript is related to an interesting application on mild technology to improve the quality of stabilized orange juices. In the introduction where the authors describe the thermal treatments could be applied should remind the influence of the pH since if we are generally referring to "fruit juices" and nnot only to citrus products the pH can be different and thus influencing the thermal treatment effect. The experimental design related to conventional and PEF assisted pasteurization might consider a comparison on F0 values computed from the thermal profiles and to use those values to compare the effect on quality parameters as a function of the different F0 values. It could be very useful to give an explanation of the difference in quality due to the applied thermal treatments, with and without PEF treatment 

Reply:
We would like to thank the reviewer for his/her time to review the manuscript and provide some suggestions for improvement.
We have updated the introduction and explained the impact of different pH of food products, and the effect of pH on the selection of thermal processing conditions. Different categories based on the pH are explained, and it is clear that citrus juices are belonging to the high-acid juices.
The second point that is raised by the reviewer is about calculating a F0 for the thermal profile and compare it the PEF treatment, to discriminate the thermal effect from the combined PEF+heat treatment. We really appreciated the suggestion of the reviewer and estimated PU values (pasteurisation unit, meaning the exposure of a product at 60°C for 1 minute, which is similar to F0 values). The following PU values were calculated: PU=0.76 (PEF-blanco at 45°C),  PU=0.81 (PEF 90°C), PU=1.69 (LH) and PU=2.15 (HH). Based on these calculations, but also visible from Figure 1, is that both thermal treatment at low and high temperatures have a much higher heat load than the different PEF treatments. A direct comparison between heat and PEF treatment is therefore not possible, and if we would like to compare data it will be done on extrapolations based on (only) 2 datapoints. More thermal data is required to discriminate the thermal effect from the PEF+heat effect.
However, we like the idea of the reviewer, and therefore we have added this suggestion for further research in the conclusion section.

Reviewer 2 Report

The authors piled a lot of data without extracting key points. The 'Conclusions" had 46 lines without providing "conclusions". The volatile data are the major part of the manuscript, however, the authors did not present the results and even without mention in the abstract. 

Author Response

The authors piled a lot of data without extracting key points. The 'Conclusions" had 46 lines without providing "conclusions". The volatile data are the major part of the manuscript, however, the authors did not present the results and even without mention in the abstract. 

Reply:
We would like to thank the reviewer for his/her time to review the manuscript.
The reviewer is not very specific in his/her comment and we do not agree that it is merely a lot of data without conclusions. In our view, we do go beyond the data and do provide conclusions. Apparently, the reviewer has another perception of conclusions than we have. What we do pick up from this criticism is that we can indeed spend a few more words in the abstract on the fact that volatile data have been obtained, although it was mentioned that there were differences in flavor compounds. We have updated the abstract and the conclusions.
With respect to the presentation of the results; all raw data of the volatile measurements are provided in Appendix A, and they are summarized in Table 2 and described in the main text.

Reviewer 3 Report

Generally, the article "Effect of Pasteurization by Moderate Intensity Pulsed Electric Fields (PEF) Treatment Compared to Thermal Treatment on Quality Attributes of Fresh Orange Juice" is within the scope of the journal of Foods. The manuscript (MS) contains some innovations which may be useful for the food industry. The MS writing quality is excellent and well organized, and the data is adequately presented for most parts. However, some questions need to be revised, as follow bellow:

1. Abstract:

Please revise the abstract, at the end of the abstract, the authors should add the necessity for some further works, to elucidate other questions of PEF in Fresh Orange Juice, if any.

2. Introduction

I think that can be positive for the article if the authors talk about the general concept of "emerging technologies", and cite other technologies, for example (ohmic heating, cold plasma, and others). The follow articles can be cited: 

https://doi.org/10.1016/j.foodres.2022.111827 

https://doi.org/10.1016/j.foodres.2021.110479

3. Figures 2, 3, 4, and 6

The bars on the figures can have different draws, in case that the articles are printed without color, for example. Example: lines inside the bar corresponding to Juice A, can be drawn left, and the lines of Juice B, to the right. And please, add legends that can be easy to identify the samples.

4. References and citations

The mainly critical point of the manuscript is the very old references. The MS has 40 references, and just one has less than five years. I suggest updating the references in the text.

Author Response

Comments and Suggestions for Authors

Generally, the article "Effect of Pasteurization by Moderate Intensity Pulsed Electric Fields (PEF) Treatment Compared to Thermal Treatment on Quality Attributes of Fresh Orange Juice" is within the scope of the journal of Foods. The manuscript (MS) contains some innovations which may be useful for the food industry. The MS writing quality is excellent and well organized, and the data is adequately presented for most parts. However, some questions need to be revised, as follow bellow:

Reply:
We would like to thank the reviewer for his/her time to review the manuscript and suggestions to improve the manuscript.

  1. Abstract:

Please revise the abstract, at the end of the abstract, the authors should add the necessity for some further works, to elucidate other questions of PEF in Fresh Orange Juice, if any.

Response 1: we have revised the abstract

  1. Introduction

I think that can be positive for the article if the authors talk about the general concept of "emerging technologies", and cite other technologies, for example (ohmic heating, cold plasma, and others). The follow articles can be cited: 

https://doi.org/10.1016/j.foodres.2022.111827 
https://doi.org/10.1016/j.foodres.2021.110479

Response 2: we have revised the section in the introduction and better explained the general concept of emerging technologies. Also other technologies are added, including citations.

  1. Figures 2, 3, 4, and 6

The bars on the figures can have different draws, in case that the articles are printed without color, for example. Example: lines inside the bar corresponding to Juice A, can be drawn left, and the lines of Juice B, to the right. And please, add legends that can be easy to identify the samples. 

Response 3: We have checked the indicated Figures, and evaluated the suggestion of the reviewer. Putting the lines from juice A to the left and those of juice B to the right would increase the number of variations to fill the bars and making it even less readable instead of  improving it. As the journal encouraged us to use colours instead of black-white taints, we have carefully selected these colours to make it easy readable for people that are colour-blind, while there is still a clear distinction between the bars when printing it black-white. Legends were not added, as it can be read from the figure caption what treatment was indicated, and the same structure is used in the graphs untreated at left, PEF control and PEF treatment in the middle and thermal treatment at the right.

Figures in the appendix were improved by adding the indication ‘PEF’ to the PEF treated samples.

  1. References and citations

The mainly critical point of the manuscript is the very old references. The MS has 40 references, and just one has less than five years. I suggest updating the references in the text.

Response 4: we have updated the text with new references.

Round 2

Reviewer 1 Report

I consider the improvement the authors have done is sufficient to allow the publication of this manuscript

Author Response

thank you very much.

Reviewer 2 Report

The authors properly addressed the reviewers' comments, and it is publishable now.

Author Response

thank you very much

Reviewer 3 Report

Dear authors,

Thanks for responding my considerations. Your improvements on the abstract and introduction sections improved the manuscript. About the figures, I understand your explanation, and I agree with them.

So, the last and critical point is that it is essential to consider the references again. As you can see, many references are old, and the update you made is insufficient, mainly on Material and Methods, and Discussion. I think the manuscript is excellent and has good results for the food industry and science community. Therefore, the methods and discussion of results must be based on actual literature. Please, provide a new update of references. 

Author Response

Dear reviewer,

thank you for your input. With respect to the last suggestion that remains, we have undertaken some action, with the following result.
We searched via “ScienceDirect” to see if there was any literature in the field of “orange juice” in the period 2018—2023, and we screened all titles for relevance to our manuscript.

Although a lot of articles have published in this field (for example on focusing on using pulsed electric fields to enhance extraction yield in orange juice, microbial inactivation in orange juice), there were only a few titles that might be of interest, and we’ve screened those articles in more detail. However, to the best of our knowledge, we think there isn’t anything that could be relevant to add to our paper.

For example, one of the potentially interesting articles entitled “a review of pectin methylesterase inactivation in citrus juice during pasteurization” published in 2018 refers to exactly the same sources as we do to indicate how PME can be measured and what limits are used to determine if a juice is shelf stable or not. We are aware that these are relatively old references, but we found no updates during the last years. When we contacted a large industrial orange juice processing company, they mentioned that they are referring to these old references too. In this review of 2018, also the effect of different PEF treatments with variable conditions on PME inactivation were reviewed, but all included PEF treatments are associated with a lot of heat that was generated as a result of the pulsing, and therefore pulse and thermal effects could not be distinguished from each other. 

In our manuscript, we tried to refer to only the original studies or to studies that are relevant to our article. Some recently published studies refers to the same references as we used, but the article does not contain any updated or new element to make it relevant to refer to them as well. For example, untreated, HPP and thermal pasteurisation of orange juice has been compared and authors see that thermal treated juice gives a slightly browner color in L*a*b* values compared to untreated and HPP. As far as we are concerned, this is not relevant to add in our M&M or Discussion, especially because it is only about 1 thermal treatment which is different from the conditions we used. Therefore, we did not add it in our reference list.

All in all, to the best of our knowledge we could not add newer references to our manuscript as they were not adding anything that is relevant to mention to our story. We hope that you understand this and that you can accept the manuscript for publication.

On behalf of the authors,
Rian Timmermans